# Host Defenses to Viruses: Lessons from Inborn Errors of Immunity

**DOI:** 10.3390/medicina58020248

**Published:** 2022-02-07

**Authors:** Lucia Leonardi, Beatrice Rivalta, Fabrizio Leone, Caterina Cancrini, Carlo Caffarelli, Gian Luigi Marseglia, Fabio Cardinale

**Affiliations:** 1Maternal, Infantile and Urological Sciences Department, Sapienza University of Rome, 00161 Rome, Italy; lucia.leonardi@uniroma1.it (L.L.); fabrizio.leone@opbg.net (F.L.); 2Research Unit of Primary Immunodeficiencies, Academic Department of Pediatrics, Immune and Infectious Diseases Division, Bambino Gesuù Children’s Hospital, IRCCS, 00165 Rome, Italy; beatricerivalta@gmail.com (B.R.); cancrini@med.uniroma2.it (C.C.); 3Department of Systems Medicine, University of Rome “Tor Vergata”, 00133 Rome, Italy; 4Pediatric Clinic, Department of Medicine and Surgery, University of Parma, 43121 Parma, Italy; carlo.caffarelli@gmail.com; 5Pediatric Clinic, Fondazione IRCSS Policlinico San Matteo, 27100 Pavia, Italy; gl.marseglia@smatteo.pv.it; 6Department of Pediatrics, University of Bari, Azienda Ospedaliera-Universitaria “Consorziale-Policlinico,” Ospedale Pediatrico Giovanni XXIII, 70123 Bari, Italy

**Keywords:** viruses, host, immunity, genetic, susceptibility, EBV, HSV, influenza, SARS-CoV-2

## Abstract

The constant battle between viruses and their hosts leads to their reciprocal evolution. Viruses regularly develop survival strategies against host immunity, while their ability to replicate and disseminate is countered by the antiviral defense mechanisms that host mount. Although most viral infections are generally controlled by the host’s immune system, some viruses do cause overt damage to the host. The outcome can vary widely depending on the properties of the infecting virus and the circumstances of infection but also depends on several factors controlled by the host, including host genetic susceptibility to viral infections. In this narrative review, we provide a brief overview of host immunity to viruses and immune-evasion strategies developed by viruses. Moreover, we focus on inborn errors of immunity, these being considered a model for studying host response mechanisms to viruses. We finally report exemplary inborn errors of both the innate and adaptive immune systems that highlight the role of proteins involved in the control of viral infections.

## 1. Introduction

Humans and viruses have been coevolving for millennia to survive each other [1]. Viruses are obligate intracellular parasites, requiring host synthesis mechanisms to achieve their replication and therefore to replicate and propagate to other cells and hosts. The ability of viruses to replicate and disseminate is countered by the antiviral defense mechanisms that the host mounts. Precisely, the ability of a given virus to productively infect a specific cell, tissue or host species is known as tropism and is due to various factors, including specific viral recognition by receptors expressed by the host [2]. Acute cytolytic infection is the most common virus–host cell interaction, characterized, when not abortive, by viral reproduction resulting in the destruction of the infected cell, followed by viremia and virus dissemination to other cells/tissues. These viruses overcome the immune response by infecting new hosts and rarely persist. Some viruses can instead establish a chronic infection by successfully entering cells and integrating them into the host genome, while not undergoing productive replication until being reactivated (latent or persistent infection) (Figure 1) [2]. Although most viral infections are generally controlled by the non-immunosuppressed host’s immune system, resulting in mild clinical manifestations, it is essential to point out that the same virus can be responsible for a different clinical phenotype in different subjects. Indeed, for most viral infections, the outcome can vary widely depending on viral factors or host features, including the structure of the virus (RNA virus versus DNA), the viral load, the site of entry in the host, the infected cell types and viral immune-evasion strategies. Many modifiable factors potentially impacting host immune defenses have also been reported, including sleep [3], nutritional status, microbiota and even psychophysical factors (Figure 2) [4,5]. Among non-modifiable factors, age and genetic susceptibility might be considered the most important. Indeed, studies on immune system ontogenesis highlighted how both young and elderly populations present with impaired pro-inflammatory and predominant Th2 responses following viral infections [6]. Epidemiological data confirm this specific susceptibility to infections in the two extreme age groups [7]. A considerably broad number of studies have also shown how genetic susceptibility may impact viral infection frequency and severity [8]. Here we provide a brief overview of information critical to understanding host immunity to viruses and immune-evasion strategies developed by viruses.

## 2. Host Immunity

Immunity to viral infection is due to specific and nonspecific mechanisms that seek out and destroy both free viruses and infected host cells. The first line of defense against any pathogen is given by the integrity of the anatomical barriers (skin, antimicrobial peptides, pH of the mucous membranes, etc.) and by antiviral restriction factors, these being host cellular proteins that are able to block viral replication and propagation (APOBEC3G, SAMHD1, TRIM5α, etc.).

When a virus can overcome the host’s first line of defense, the infection begins. Innate immunity is mainly characterized by the response of local sentinel immune cells, the production of type III and I interferon and the killing of infected cells by natural killer (NK) cells (Figure 3) [9]. Sentinel cells of innate immunity are activated a few hours from the infection by the recognition of patterns of molecules uniquely expressed by viruses (pathogen-associated molecular patterns (PAMPs)) by specific host cell receptors (PRRs).

The innate response can stop the infection or prevent virus dissemination while promoting the activation of the adaptive response. When the innate immunity mechanisms do not adequately counteract the infection, a cell-mediated adaptive response intervenes in less than a week. Activated cytotoxic T lymphocytes eliminate infected cells by recognizing viral peptides expressed on cell surfaces on class MHC I molecules. In addition, the adaptive humoral response produces neutralizing antibodies that are able to bind viruses during viremia, preventing virus dissemination to target cells. Moreover, being responsible for opsonization and subsequent phagocytosis, antibodies participate in the elimination of virally infected cells. Nevertheless, patients with profound B cell immunodeficiency present with frequent and severe bacterial infections and are less prone to viral infections [10].

Indeed, in the kinetics of the immune response to viruses, the activation of the interferon system mentioned above seems to be the most essential antiviral mechanism, as it aims at containing the infection in its initial stages [11]. Of note, the term “interferon” derives from the ability of this cytokine to interfere with the development of viral infection. IFN I production is the result of the recognition of specific viral PAMPs by host endosomal or cytoplasmic PRRs that activate intracellular signaling pathways and consequently transcription factors, such as IRF3, in virus-infected cells, and IRF 7, in pDCs (plasmacytoid dendritic cells) that can produce interferon even if not infected. IRF3 and IRF7 translocate into the nucleus and activate the transcription of type I interferon (IFNα and IFNβ). Circulating IFN will therefore activate NK cells and cytotoxic lymphocytes killing. Moreover, both IFNα and IFNβ activate the transcription of ISGs (interferon-stimulated genes), responsible for inhibiting viral replication in infected cells and viral dissemination (Figure 4) [11].

## 3. Virus Strategies of Immune Evasion

Viruses have developed numerous effective immune evasion strategies to survive host defenses, including blocking antigen presentation, escaping from apoptosis and cytokine production, evading NK cells activities and changing their antigen expression [12].

Viruses can modify their surface antigens, ceasing to be recognizable by the immune cells. These antigenic modifications occur by point mutations, in which the resulting virus is still partially recognizable by the host (antigenic drift), or by recombination of the genes of a virus with those of correlated viruses, with consequent significant changes in the resulting protein structure that becomes less or not recognizable by antibodies and T cells directed against the previous variant (antigenic shift).

Moreover, latency and persistence are considered effective viral evasion mechanisms from host’s defenses.

In latent infections, the virus persists in the cell without replicating and killing the infected cell, as in the case of herpes viruses. In the case of immunosuppression, the latent infection may reactivate with a consequent new spread of the virus. As a consequence, an inapparent nonlytic infection may turn into a severe, either fulminant or chronic–active disease.

In persistent infection, the virus is responsible for a chronic infection that can result in host tissue damage and/or hyperinflammatory conditions, such as macrophage activation syndrome (MAS) or polyclonal lymphoid proliferations.

Many additional mechanisms of evasion that viruses have learned to inhibit host immune responses have been described, including the expression of genes that code for homologs of the receptors of some cytokines [13].

## 4. Viruses and Inborn Errors of Immunity

It is now known that severe infections may be the result of pathogen virulence but also of possible genetic defects impairing host immunity.

Inborn errors of immunity (IEIs) represent a large and heterogeneous group of monogenic disorders characterized by immune cells’ development or functional impairment, leading to a variable susceptibility to infections. To date, more than 430 inborn errors of both innate and adaptive immunity have been molecularly characterized [14]. Susceptibility to severe viral diseases is reported in IEIs of both innate and/or adaptive immune responses. These patients may develop life-threatening viral infections. These infections may be severe, persistent, recurrent or refractory to therapy, and could also be caused by low virulence viruses.

In the past 70 years, studies of patients with severe viral infections suspected of having IEIs and their long-term follow-up have clarified several aspects of viral infection pathogenesis and antiviral immune mechanisms. First of all, the evidence for increased susceptibility to infections as the predominant manifestation of almost all forms of IEIs confirmed the paramount importance of the immune system in defense against microbes. Subsequently, a variable susceptibility of different immune defects to the different infectious etiologies (bacteria, fungi or virus) has been progressively described [15]. Molecular and cellular IEI characterization has changed our understanding of pathogenesis by revealing previously unknown host defense mechanisms, while disproving immunological notions initially suggested by studies on animal models. Specifically, clinical studies, conducted on patients affected by IEIs associated with viral infections, highlighted how only some mechanisms are essential, while others are redundant or uninvolved in antiviral responses [16].

Some IEIs predispose to multiple viral and/or even multiple microbial infections (Figure 5), whereas other IEIs, mostly described in the last years, are characterized by a specific genetic susceptibility to a restricted number of pathogens or viruses (Table 1).

## 5. IEIs Characterized by Severe Infections with Multiple Viruses

The prototypic example of an IEI associated with opportunistic viral infections is SCID (severe combined immunodeficiency), characterized by profound alterations of T lymphocytes and a variable absence of B and NK cells, for example, X-linked SCID (g-chain), JAK3, IL7Ra, RAG1/2, DCLRE1C/Artemis, ADA. Early onset in the first months of life and life-threatening infections caused by a broad range of pathogens (bacteria, viruses and fungi) are the main clinical features of these conditions. Studies of SCID patients were crucial in elucidating the role of T-cell immunity in controlling intracellular pathogens by eliminating virally infected cells. [27]

In some cases of atypical/leaky SCID or CID (combined immunodeficiency), where T cell function is impaired but not abolished, viral infections (i.e., Ebstein–Barr virus (EBV) or CMV)) may trigger other complications, such as immune-dysregulation manifestations (cytopenia, lymphoproliferation and hyperinflammation). An example is RAG deficiency, where the hypomorphic mutations of RAG1 or RAG2 genes, necessary for the recombination of T and B receptors, lead to the proliferation of oligoclonal/autoreactive T and B cells, frequently sustained by a viral infection, such as CMV, and responsible for immune dysregulation manifestations [19,28].

Viral infections may also be responsible for the onset of malignancies in some IEIs. Different mechanisms may explain the pathogenesis of malignancies, including virus latency, integration in the human cell genome or the persistent trigger of B cell proliferation [29,30].

An example of CID where viral infections contribute to immune dysregulation and malignancy onset is APDS (activated PI3K-kinase delta syndrome). APDS patients present with lymphopenia, hypogammaglobulinemia and hyper IgM, recurrent sinopulmonary infections, chronic herpes virus infections (i.e., EBV) and immune dysregulation manifestations. The latter include cytopenia, arthritis, inflammatory bowel diseases (IBD) and an intrinsic predisposition to persistent non-neoplastic splenomegaly/lymphoproliferation and lymphoma. The constitutive activation of the PI3K-Akt–mTOR pathway leads to impaired activity of CD8+ and NK cells exhibiting an immunosenescent/exhausted phenotype and to the expansion of transitional B cells which may act as virus reservoirs. Moreover, the altered antibody response, unable to control the spread of the virus, may contribute to viral persistence. The impaired EBV clearance and the intrinsic B cell proliferation together contribute to lymphoma pathogenesis [20,31,32,33].

## 6. IEIs Characterized by Unique Predisposition to a Single Virus

Some IEIs are characterized by a specific susceptibility to a narrow range of pathogens. This selective susceptibility demonstrates that there are redundant mechanisms of protection for some groups of pathogens. Thus, although the impaired pathways are involved in the response to multiple pathogens, some proteins can be crucial for the control of a single pathogen and/or in specific tissues [34]. The study of these IEIs and the identification of the associated impaired pathways allow us to understand and distinguish the different mechanisms involved in host defense and eventually to identify a target for specific therapies and vaccines.

### 6.1. IEIs with a Specific Susceptibility for EBV

Some IEIs are characterized by a particular susceptibility for EBV infection [21,22]. Patients with these conditions may develop severe, acute or chronic complications, such as hemophagocytic lymph histiocytosis (HLH) or lymphoproliferative disorders.

More than 90% of the population has been exposed to EBV, a human herpes virus, capable of remaining latent in host B cells. The clinical manifestations are associated with virus latency and the primary infections depend on the age and immunocompetence of the human host.

Usually when the primary EBV infection develops in a healthy child is asymptomatic or paucisymptomatic whereas in young adults is frequently associated with acute systemic symptoms (such as fever, pharyngeal pain, hepatosplenomegaly and adenomegalies). Symptoms depend on viral load, the type of HLA (human leukocyte antigen) that influences the immune repertoire’s variety [35] and the efficacy of effector and regulatory mechanisms of innate immunity (mostly NK and T gamma–delta) variable across the lifespan. Innate immunity (NK, NKT, T gamma–delta) is the first arm of the immune system that controls EBV infection. Subsequently, adaptive immunity, most of all CD8+ T cytotoxic cells, has a crucial role in controlling the primary and the latent state of EBV infection [22,36,37].

Some IEIs (including XLP1 andXLP2) are characterized by a defective cytotoxic killing of EBV-infected B cells and are associated with protracted T and NK expansion and activation, leading to histiocyte proliferation, cytokine overproduction, hyper inflammation and hemophagocytosis (HLH). XLP1 is caused by a mutation of SAP, a protein involved in the activation of NK and CD8+ by B cells. XLP2 is due to a mutation of the XIAP protein, which is involved in apoptosis and the survival of cells. In XLP2, although the mechanism of EBV-driven HLH is still unclear, the increased death of T cells after activation and impairment of their effector function probably leads to the persistence of EBV-infected B cells and triggers hyperinflammation [20,30,32]. IEIs characterized by a defect in granule-dependent cytotoxicity of NK cells and CD8+ lymphocytes necessary to eliminate the virus induce a persistent inflammatory trigger leading to severe HLH. Among these perforin/PERF deficiency, LYST, UNC13D or RAB27A [21,22].

IEIs, with increased susceptibility to EBV, characterized by the inability to eliminate proliferating EBV-infected B cells over time, lead to an increased risk for B-cell lymphoproliferative disorders and lymphomas sometimes associated with HLH. Among these IEIs are included conditions with an impairment in T-cell activation (such as RASGRP1, MAGT1 and ITK deficiencies), defects of DNA metabolism essential for the expansion of activated antigen-specific T cells (CTPS1 deficiency) or defects of co-stimulatory pathways (CD70, CD27 and TNFSFR9) [22].

### 6.2. IEIs Associated with a Specific Susceptibility for HSV, Influenza A and SARS-CoV-2

In the last few decades, the increasingly sophisticated characterization (next generation sequencing, etc.) of IEIs has made it possible to discover new immunological pathways that predispose to severe viral diseases involving the Toll-like receptor–interferon signaling pathways. These IEIs have been defined as non-conventional PID [16].

These studies pointed out the importance of human IFNs in protective immunity against viruses. In particular showed that while type I IFNs are fundamental for resistance to viral infections, type II IFNs are not critical in antiviral defense, being largely redundant for immunity against viruses in the natural course of infections. Finally, type III interferons (INF-λ) are critical in localized infection control at mucosal barrier sites [9].

Viral nucleic acids induce the synthesis of IFNs type I through specific host endosomal and cell surface receptors, including RIG-I, TLR 3,7,8 and 9. As a result, the transcription of the interferon regulatory factor (IRF) family is activated, leading to the expression of IFNs type I in the infected cell. This acts in a paracrine way by conferring an antiviral state to neighboring target cells through activation of the JAK–STAT pathway [9]. Congenital defects involving interferon function have shown that this pathway has a crucial role in the early activation of immunity and in changing the course of disease. Severe viral infections, such as herpesviruses, influenza A and SARS-CoV-2 infections, may result from single gene IEI of the IFN pathway. These IEIs are extremely rare, but their knowledge has allowed understanding the importance of these pathways in immunity against viruses, suggesting therapeutic strategies that have been potentially helpful in treating severe viral illness.

#### 6.2.1. IEIs with a Specific Susceptibility for Influenza A

We have known about the influenza A virus for about 80 years now and many studies have focused on the pathogenicity of this virus. To date, however, we are unable to certainly clarify why this infection is dangerous only in some subjects. The most well-known risk is underlying lung disease. However, influenza can also be severe in patients with no pulmonary impairment and who are otherwise healthy. LOF (loss of function) mutations of IRF7 have been demonstrated in children experiencing life-threatening symptoms during primary influenza infection. The loss of function of IRF7 does not allow the production of type I IFNs in response to the virus. This specific deficit, albeit rare, may be just the tip of the iceberg of specific inborn defects that can explain severe symptoms in some subjects in the course of common infections (Figure 6b) [23,24]. This makes it mandatory to further study genetics in patients presenting with abnormal symptoms to common viral infections in order to prevent, through targeted therapy, fatal complications in perfectly healthy patients, as well as further explore potential antiviral immunological pathways still unknown.

#### 6.2.2. IEIs with a Specific Susceptibility for HSV

HSV-1 is notoriously known for its ability to establish acute and latent infections evading host immunity. After recognizing the pathogen-associated molecular patterns (PAMPs) through PRRs (TLRs (especially TLR3, TLR7/8, TLR9), RIG, nod-like receptors (NODs)), the signaling pathways of interferon and its transcription factors are activated. In recent years, the function of viral IFN receptors has been widely described through genomic studies of children with isolated herpes simplex encephalitis (HSE) that have revealed mutations affecting the TL3 signaling pathways; this receptor can recognize viral dsRNA (Figure 6a). In addition, it has been shown that there is an impairment of the intrinsic antiviral activity of neurons and oligodendrocytes in these patients. This specific defect could explain why these children present with encephalitis and support reconsideration of the postulate that immune function could be limited to hematopoietic cells [23,24].

#### 6.2.3. IEIs with a Specific Susceptibility for SARS-CoV-2

In general, human coronaviruses (hCoVs) have been known as a cause of the common cold. With the first outbreak of SARS (severe acute respiratory syndrome), the first HCoV-related severe disease, in 2002, our consideration of these viruses has radically changed [38].

SARS-CoV-2, the etiological agent of the ongoing COVID-19 pandemic, was first recognized in December 2019 [39]. It replicates to very high titers in the upper respiratory tract, especially in the pre-symptomatic phase of the infection, causing in most cases mild symptoms but thereby obtaining the possibility of spreading to a greater number of subjects [40]. To date, more than 250 million people in the world have been infected, demonstrating that older people and people with chronic diseases develop more severe symptoms [41]. However, although SARS-CoV-2 is generally more virulent in older patients, male and with specific comorbidity, this group of subjects may present with clinical manifestations ranging from potentially fatal to asymptomatic infection. On the contrary, cases of young and previously healthy patients presenting with severe COVID-19 disease have been described, suggesting potential specific genetic susceptibility to SARS-CoV-2 in these subjects. During Sars-CoV-2 infection, innate defense is immediately active to limit the spread of the virus in the lungs. The recognition of viral components induces the response of IFN-I and the cytotoxic action of natural killer (NK) lymphocytes in the airways [42]. A multicenter study conducted on COVID-19 patients with severe disease highlighted that more than 3% of these subjects, previously healthy, presented with innate errors of IFN type I TLR3 and the IRF-7 dependent pathway (Figure 6c). As a result, these patients had low serum type I IFN [25,26]. In previously healthy patients presenting with a COVID-19 life-threatening infection, neutralizing autoantibodies against type I IFNs (especially INFω and INFα) have been detected [43]. These findings were also previously described in patients with IEIs and severe viral infections [44,45]. These findings suggest that the administration of type I IFN may be of therapeutic benefit in these patients and that there could be mutations in other genes related to the type I interferon in other patients presenting with severe COVID-19 infection.

## 7. Conclusions

Over millions of years, viruses have developed efficient survival strategies against host immunity. Increased susceptibility to viral infections has been described in patients affected by IEIs and these represent a model for studying host response mechanisms to viruses.

In the last few decades, the increasingly sophisticated characterization of IEIs has made it possible to identify defects in specific pathways (including the IFN pathway) that sometimes cannot be diagnosed with standard immunological analysis.

These studies have also made it possible to understand how many immunological circuits are largely redundant but essential only for immunity against one or a few specific infections and finally confirm that the immune response to infections is not a prerogative of hematopoietic cells but requires the action of different cell types.

Future studies reporting additional mutations in proteins of both innate and adaptive immune systems will highlight the role of other proteins involved in the control of severe viral infections.

## Figures and Tables

**Figure 1 medicina-58-00248-f001:**
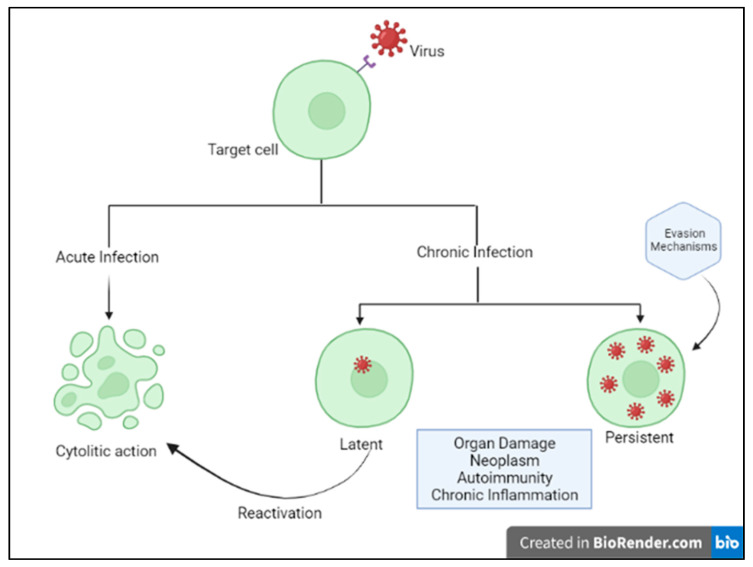
Host–virus interactions. Adapted from Immunology IV: Clinical Applications in Health and Disease by Joseph A. Bellanti.

**Figure 2 medicina-58-00248-f002:**
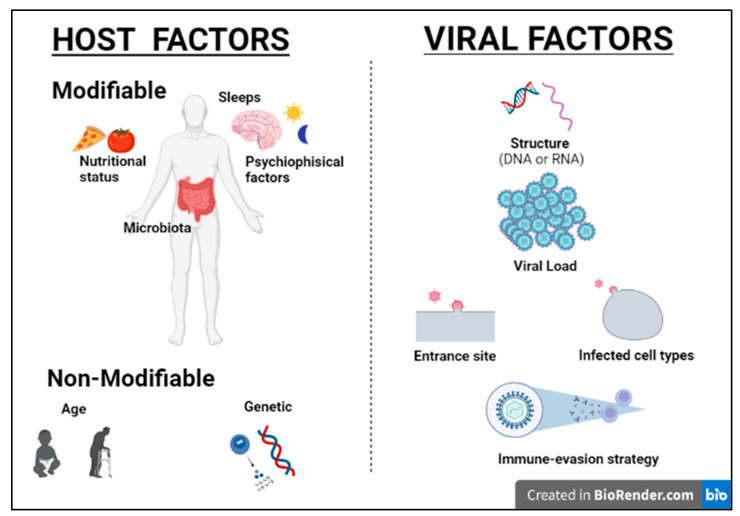
Host and viral factors impacting infection outcome.

**Figure 3 medicina-58-00248-f003:**
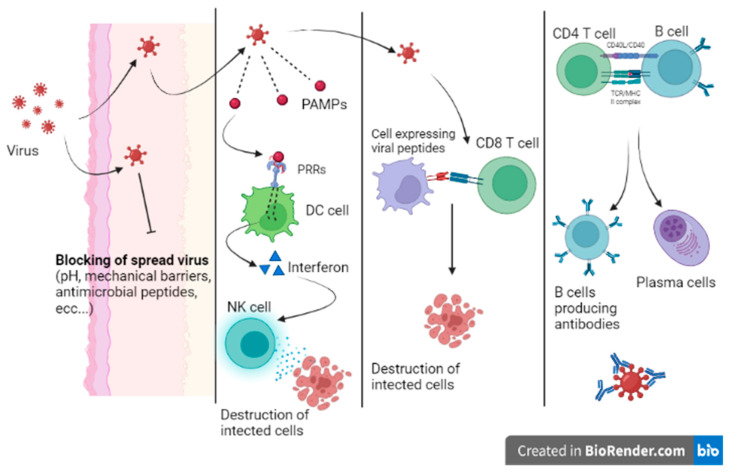
Host immunity.

**Figure 4 medicina-58-00248-f004:**
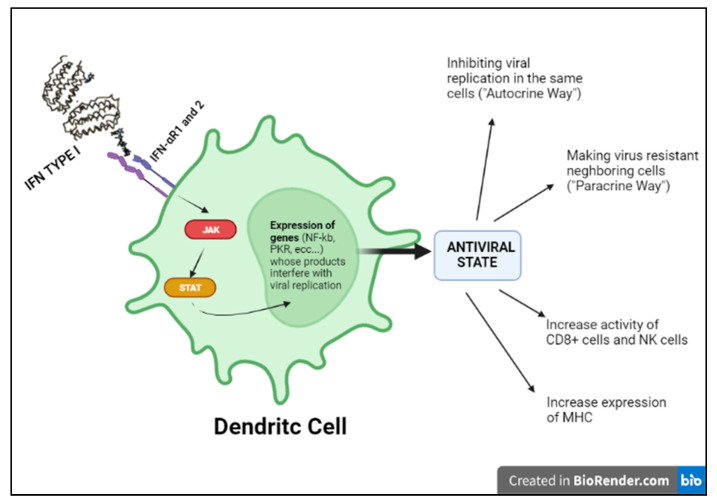
Type I INF pathway.

**Figure 5 medicina-58-00248-f005:**
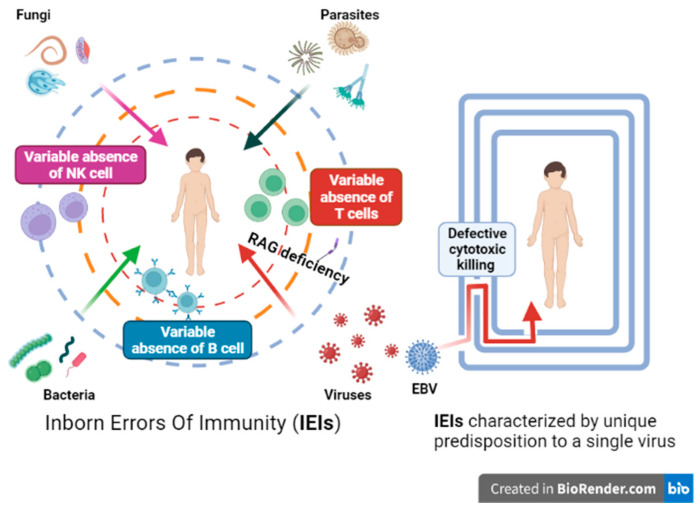
Inborn errors of immunity characterized by predisposition to a multiple or single virus (e.g., RAG deficiency impair B and T function with a susceptibility for all pathogens) and IEIs characterized by a unique predisposition to a single virus.

**Figure 6 medicina-58-00248-f006:**
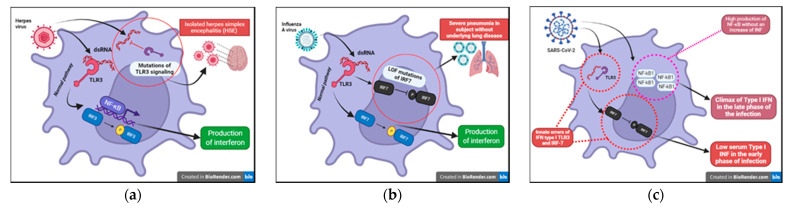
(**a**) Inborn errors of immunity conferring predisposition to childhood HSE (Herpes Simplex Encephalitis). (**b**) LOF (loss of function) mutations of IRF7 exposes children to a greater risk from severe Influenza A disease. (**c**) Innate errors of IFN type I: TL3 and the IRF-7 dependent pathway.

**Table 1 medicina-58-00248-t001:** IEIs and viral susceptibility: examples and typical presentations.

IEIs Characterized by Severe Infections with Multiple Viruses			
Disorder Group	Examples of Specific Disorder	Typical Presentation	Ref
SCID	X-linked SCID(g-chain), JAK3, IL7Ra, RAG1/2, DCLRE1C/Artemis, ADA	Failure to thrive, severe infection (also by fungi and bacteria), thrush, chronic diarrhea, rashes	[17]
Atypical SCID or CID	Hypomorfic RAG1/2, APDS	Immunodysregulation manifestations, neoplasia	[18,19,20]
**IEIs Characterized by Unique Predisposition to a Single Virus**			
**Virus**	**Example of Specific Disorder**	**Typical Presentation**	**Ref**
EBV	XLP1/2, perforin/PERF deficiency, LYST, UNC13D (Munc13-4) or RAB27A	HLH and hyperinflammation	[21,22]
	RASGRP1, MAGT1 and ITK deficiencies, CTPS1 deficiency, CD70, CD27, TNFSFR9	Lymphoproliferation and lymphoma	
Influenza A	IRF7	Life-threatening ARDS	[23,24]
HSV	TLR3	Herpes simplex virus encephalitis	[23,24]
SARS-CoV-2	TLR3, IRF7	Life-threatening ARDS	[25,26]

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
