# Peer review of "Host Defenses to Viruses: Lessons from Inborn Errors of Immunity"

_medicina, 2022, doi:10.3390/medicina58020248_

Round 1

Reviewer 1 Report

This review by Leonardi and collegues summarizes some aspects of inborn errors of immunity concerning their for in facilitating viral infections. In a first part of the review, the authors explain basic aspects of the immune response against viruses, while the second part highlights some inborn errors of immunity in conjunction with specific viral infections. While this review can be regarded as a useful overview for people new to the field, it cannot serve as a comprehensive review for specialists. Nevertheless, the review is well written (in general) and helps newcomers to get oriented in the field.

I have the following comments for further improvement:

  1. It would be helpful to include a table that summarizes the most important inborn errors of immunity in conjunction with viral infections and important citations
  2. Line 39: the authors should modify this sentence; host tropism may be determined by factors beynd receptor recognition.
  3. Line 70-71: first line defense also includes the action of cell-intrinsic restriction factors. This is a very important research field that should be acknowledged.
  4. Line 97: Interferon is abbreviated with INF; I think IFN is usually used as abbreviation for interferon.
  5. Figure 4 contains a typo: “neighbouring”
  6. Line 128: the definition of persistent infection as hiding integrated into the genome is not correct. Persistent also includes infections with low level continuous production of viruses (e.g. HCV).
  7. Line 143-144: I am not sure whether this statement is correct. Since inborn errors of immunity are rare events, their effects are only recognized in the context of very frequent infections like EBV, CMV. However, with SARS-CoV-2, we now have a very frequent infection with rather high virulence where we can also see the effects of inborn errors of immunity. I am quite sure that similar effects could also be observed with other high virulence viruses.
  8. Lines 178-179: this sentence is not correct or at least misleading; in most cases of EBV caused malignancies no integration into the human genome can be detected; rather, the genome persists in an episomal form; “malignant degeneration” is also awkward.
  9. The manuscript contains several typographical errors (e.g. line 255: mutations….has been) or errors of grammar that need to be corrected.

Author Response

We thank the reviewer for the valuable comments and suggestions.

  1. A table summarizing the inborn errors of immunity associated with viral infections (described in the text) and the related citations has been added
  2. Line 39: the sentence has been modified highlighting that viral tropism may be determined by factors beyond receptor recognition.
  3. Line 70-71:  the action of cell-intrinsic restriction factors as been cited among first line host's defense
  4. Line 97: Interferon has been abbreviated with IFN 
  5. Figure 4 has been modified and typo corrected
  6. line 128: we modified the definition of persistent infection 
  7.  We changed line 143-144, in this sentences we want highlight the susceptibility to infection rather than the risk for high virulence viruses.
  8. we changed Lines 178-179 including different mechanisms involved.
  9. We corrected typographical and grammar errors 

Reviewer 2 Report

The authors have made a review of the host immune responses to viruses, evasion by viruses and effect of inborn errors of immunity (IEIs) on virus pathogenesis. This paper presents latest information on the deficiencies created by IEIs and how they influence course of virus  This article will be useful for scientists studying molecular aspects of virus pathogenesis and immune responses against different infections. Corrections mentioned in the attached file may be carried out.

Author Response

We thanks the reviewer for the suggestions.

All corrections required have been done

We deleted at line 35 " the  nutritional needs"

We replaced the word "plasma cell" in figure 3 

We correct the errors :  

at line 112 changed "for with by",

at line 115 changer "became in becomes",

at line 128 changed "hiding in having", 

at line 169 added by,

at line 198 changed "pathogens in pathogen",

line 309 changed "intercome in interferon",

in Fig 6b changed "HAV in HSV",

we rephrased the sentence at line 324 - 327